# EMERGENT COVERT SIGNALING IN ADVERSARIAL REFERENCE GAMES

**Dhara Yu**[1], **Jesse Mu**[1], **Noah D. Goodman**[1,2]
Departments of Computer Science[1] and Psychology[2]
Stanford University
{dharakyu,muj,ngoodman}@stanford.edu

## ABSTRACT

Emergent communication is often studied in dyadic, fully-cooperative reference games, yet many real-world scenarios involve multiparty communication in adversarial settings. We introduce an *adversarial reference game*, where a speaker and listener must learn to generate referring expressions without leaking information to an adversary, and study the ability of emergent communication systems to learn *covert signaling* protocols on this task. We show that agents can develop covert signaling when given access to additional training time or shared knowledge over the adversary. Finally, we show that adversarial training results in the emergent languages having fewer and more polysemous messages.

## 1 INTRODUCTION

Human language is a uniquely flexible and powerful tool for human collaboration. To build models that can learn language-like communication skills, previous work has trained agents to develop language protocols from scratch to solve collaborative tasks (see Lazaridou & Baroni, 2020 for a review). One dominant setting in emergent communication is the fully-cooperative Lewis reference game (Lewis, 1969) with two agents, a speaker and a listener, in some common grounded context, where the speaker must produce a message that allows the listener to identify the correct referent. However, human language is used for much more than fully-cooperative, two-player signaling games. In particular, semi-cooperative and competitive behavior among multiple agents is the norm in a variety of everyday scenarios, such as negotiation (Nash, 1950) or covert signaling (Smaldino et al., 2018), where individuals may have both shared goals and conflicting interests. Accordingly, there is a need to study how language conventions change in more complex settings involving multiple parties, with more nuanced incentive structures.

Towards this aim, we introduce a three-player *adversarial reference game*, which involves a speaker, a listener, and a third *adversary*. The goal of the game is for the speaker to send a message that can be used by the listener to identify a referent, but that *cannot* be used by the adversary. To succeed, the speaker and listener must form a *covert signaling* protocol to communicate while avoiding leaking information to the adversary: this setting is common in real-world scenarios such as communication among political dissidents (Kuran, 1989; Coopman, 2011), or cryptographic communication (Abadi & Andersen, 2016).

By training agents to play these games, we aim to answer the following research questions: (1) Under what conditions can the speaker and listener agents develop a successful covert signaling protocol? (2) How do the properties of the emergent language change under adversarial pressure? We find that a speaker and listener can develop a covert signaling protocol, but only when granted advantages over the adversary, such as additional training time or a private source of common ground. We also examine how the languages change as a result of adversarial pressure: we suggest that speakers and listeners obscure their language by sending fewer yet more polysemous messages, which degrades the ability of agents to communicate about out-of-domain concepts.

## 2  THE ADVERSARIAL REFERENCE GAME

Formally, an *adversarial reference game* consists of 3 players: a speaker $S$, a listener $L$, and an adversarial listener $A$, each parameterized by $\theta_S, \theta_L$ and $\theta_A$, respectively. We say that $S$ and $L$ form a *coalition* because that they have the shared incentive to communicate while avoiding information leakage to the adversary.

Each game involves $N$ inputs $\{i_1, \ldots, i_N\}$, with one input designated as the target $t \in [1, N]$, revealed only to the speaker $S$. The role of $S$ is to produce a message $\boldsymbol{m}$ given the inputs and target: $S(\boldsymbol{m} \mid i_1, \ldots, i_N, t; \theta_S)$, where $\boldsymbol{m}$ is some discrete sequence of tokens $\boldsymbol{m} = (m_1, \ldots, m_k)$ with $m_i \in V$ (a fixed vocabulary) and $k \leq K$ (some maximum message length).

The goal of both the listener $L$ and the adversary $A$ is to identify the correct target input given the speaker's message: $L(\hat{t} \mid i_1, \ldots, i_N, \boldsymbol{m}; \theta_L)$ and $A(\hat{t} \mid i_1, \ldots, i_N, \boldsymbol{m}; \theta_A)$, respectively. The key difference is how $L$ and $A$ are trained. $A$ is trained independently of $S$ and $L$ to maximize the likelihood of guessing the target given the message $\boldsymbol{m}$ by minimizing the cross-entropy loss:

$$\mathcal{L}_A(\theta_A) = \mathbb{E}_{\boldsymbol{m} \sim S}\left[-\log A(t \mid i_1, \ldots, i_N, \boldsymbol{m}; \theta_A)\right]. \tag{1}$$

$S$ and $L$ are trained so that the speaker generates messages $\boldsymbol{m}$ that maximize success of $L$ *and minimize success of* $A$. They jointly optimize the loss function

$$\mathcal{L}_{S,L}(\theta_S, \theta_L) = \mathbb{E}_{\boldsymbol{m} \sim S}\Big[\underbrace{-\log L(t \mid i_1, \ldots, i_N, \boldsymbol{m}; \theta_L)}_{\text{Listener surprisal}} + \underbrace{\log A(t \mid i_1, \ldots, i_N, \boldsymbol{m}; \theta_A)}_{\text{Negative adversary surprisal}}\Big]. \tag{2}$$

Note that the coalition cannot modify the adversary parameters, and vice versa. In other words, $\theta_S$ and $\theta_L$ are held constant when updating $\theta_A$, and $\theta_A$ is held constant when updating $\theta_S$ and $\theta_L$.

This setup bears similarity to cryptographic encryption with neural networks (Abadi & Andersen, 2016); the key difference is that the goal of encryption is to provide a general-purpose approach for secure information exchange, whereas our goal is to induce a discrete covert signaling protocol, and analyze the resulting languages in line with similar work in emergent communication.

In early experiments, we found that jointly optimizing $\arg\min_{\theta_S, \theta_A, \theta_L} \mathcal{L}_A(\theta_A) + \mathcal{L}_{S,L}(\theta_S, \theta_L)$ was unable to induce covert signaling. Instead, similar to Abadi & Andersen (2016), we adopt an alternating optimization procedure, where $S$ and $L$ are trained for $r$ steps, then $A$ is trained for 1 step. Across experimental conditions, we vary the value of $r$, where a larger $r$ represents a larger computational advantage in favor of the coalition. Additionally, we examine the effect of introducing shared common ground between $S$ and $L$ in the form of $J$ additional private inputs $i_1^{\text{priv}}, \ldots, i_J^{\text{priv}}$. $S$ and $L$ are then updated to condition on these private inputs: $S(\boldsymbol{m} \mid i_1, \ldots, i_N, t, i_1^{\text{priv}}, \ldots, i_J^{\text{priv}}; \theta_S)$ and $L(t \mid i_1, \ldots, i_N, i_1^{\text{priv}}, \ldots, i_J^{\text{priv}}, \boldsymbol{m}; \theta_L)$, respectively.

## 3  DATASET AND EVALUATION

We use the ShapeWorld dataset (Kuhnle & Copestake, 2017), a grounded visual reasoning dataset consisting of synthetic images. Each example reference game contains 5 colored shape images over which the speaker produces a message and the listeners disambiguate. There are 10 possible colors and 9 possible shapes for a total of 90 color-shape pairs, which we refer to as *concepts*. We perform data augmentations on the images during training so that each agent sees a different "view" of the same image, to discourage overfitting of the communication protocol to low-level visual features (Dessí et al., 2021). We found that applying these augmentations increased systematicity of the speaker messages and the listener accuracy in the non-adversarial setting, which motivated us to also apply them to the adversarial game. Further details on the data are available in the Appendix.

To evaluate the effectiveness of the resulting emergent communication, we first compute **accuracy** for $L$ and $A$ on the reference games, with successful covert signaling indicated by differentially higher accuracy for $L$. We evaluate the agents' generalization ability by measuring accuracy on two test sets, with one containing only concepts seen during training, and the other containing only concepts where the concept color was unseen during training (meaning that every concept in this test set is novel to the agents). We refer to these two splits as the *seen* test set and *unseen* test set, respectively.

To measure the correlation between concepts and messages, we compute the **normalized mutual information** between messages $M$ and concepts $C$. The normalized mutual information $\text{NMI}(M, C)$ yields a value between 0 and 1, with larger values representing higher alignment between concepts and messages. We also measures the **conditional entropies** of messages and concepts $H(M \mid C)$ and $H(C \mid M)$ and count the number of **unique messages** sent across all games in the dataset.

## 4 EXPERIMENTS

Each agent is a CNN-RNN network trained end-to-end. In each agent, the images are encoded via a CNN. The speaker then uses the image embedding to condition a GRU (Cho et al., 2014), from which a message is sampled using the Gumbel-Softmax trick with the straight-through estimator (Jang et al. 2017; Maddison et al. 2017). The listeners encode the message with a GRU and compute a dot product between the image embedding and the message embedding which represent the logits over image targets (see Appendix for more details).

### 4.1 EMERGENCE OF COVERT SIGNALING

In our experiments, we manipulate two experimental variables: the *computational advantage*, and the *common ground advantage*, that the speaker and listener possess relative to the adversary.

**Computational advantage:** This experiment examines whether increasing the amount of training time to develop conventions affects task success and the emergent language. In this setting, the coalition takes more uninterrupted gradient steps compared to the adversary, i.e. increasing $r$. The lower bound ratio 1:1 indicates an equal number of gradient steps taken by the coalition and the adversary, while at the upper bound 50:1, the coalition takes 50 steps for every one taken by the adversary.[1]

**Common ground advantage:** This experiment examines whether manipulating the amount of additional common ground among the coalition affects downstream behavior. In this setting, the coalition has access to a bank of additional images containing random concepts that the adversary cannot see. The computational ratio is held constant at 1:1.

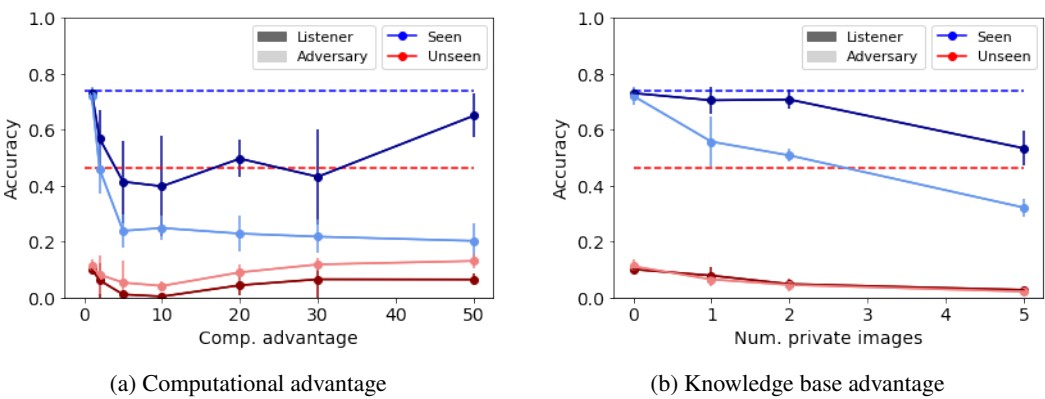

(a) Computational advantage          (b) Knowledge base advantage

Figure 1: Listener and adversary accuracy in the (a) computational advantage and (b) common ground advantage settings. Dashed lines are listener accuracy on seen (blue) and unseen (red) test sets when trained in the absence of an adversary. Points are the mean of 5 runs; error bars are stdev.

Figure 1 shows the coalition and adversary accuracy across experimental conditions. Note that since there are 5 images per game, chance accuracy is 0.2. Looking at performance on the *seen* test set in the computational advantage experiments (a), we observe that with no computational advantage

---

[1]We note that the conditions in which the computational advantage is most skewed in favor of the coalition are not simply instances of undertraining the adversary. Because we train on a simple dataset, listener accuracy in the non-adversarial setting converges quickly. Accordingly, the number of gradient steps taken by the adversary even in the most skewed setting is comparable to the number required for ceiling performance in the non-adversarial game.

| | | $\mathrm{Acc}_L$ | $\mathrm{Acc}_A$ | $\mathrm{NMI}(M,C)$ | $n$ uniq. | $H(C \mid M)$ | $H(M \mid C)$ |
|---|---|---|---|---|---|---|---|
| **True language** | | - | - | 1 | 34 | 0 | 0 |
| **Baseline (no adversarial)** | | 0.74 | 0.73 | 0.31 | 118.2 | 2.87 | 3.29 |
| **Comp. advantage** | 1:1 | 0.73 | 0.72 | 0.31 | 120 | 2.84 | 3.34 |
| | 5:1 | 0.41 | 0.24 | 0.07 | 8.4 | 5.57 | 3.58 |
| | 20:1 | 0.50 | 0.23 | 0.07 | 6.4 | 5.47 | 3.61 |
| | 50:1 | 0.65 | 0.20 | 0.11 | 12.8 | 4.73 | 3.60 |
| **Common ground** | 0img | 0.73 | 0.72 | 0.31 | 120 | 2.84 | 3.34 |
| | 1img | 0.71 | 0.56 | 0.29 | 101.4 | 3.05 | 3.35 |
| | 2img | 0.71 | 0.51 | 0.27 | 92 | 2.99 | 3.40 |
| | 5img | 0.53 | 0.32 | 0.19 | 43 | 4.17 | 3.48 |

Table 1: Effects of adversarial training on listener accuracy $\mathrm{Acc}_L$, adversary accuracy $\mathrm{Acc}_A$, normalized mutual information $\mathrm{NMI}(M,C)$, $n$ unique messages, and conditional entropies $H(C \mid M)$ and $H(M \mid C)$, on the *seen* test set. **True language** represents the compositional rules from which the dataset was derived; the number of unique messages is equal to the number of color-shape pairs. **Baseline (no adversarial)** represents agent performance and language evaluation without the adversarial loss function. Note that the 1:1 and 0img conditions are identical; they are duplicated for readability. Data points are the mean of 5 runs per condition.

(1:1 ratio), both the listener and adversary achieve high accuracy. As the computational advantage increases, both listener and adversary accuracy decrease, but adversary accuracy drops more; beyond 5:1, adversary accuracy is at chance. This indicates that the coalition has learned to evade the adversary, although at the cost of its own performance. Only when the computational advantage is heavily skewed (the ratio is 50:1) do we observe successful covert signaling: the listener returns to near baseline accuracy, while adversary accuracy stays at chance. However, in all conditions on the *unseen* test set, both listener and adversary accuracy are well below baseline levels, meaning that the adversarial pressure has led the agents to completely fail to generalize to unseen concepts.

In the common ground experiments (b), first looking at performance on the seen test set, we observe that adding 1 or 2 images as a source of common ground does not affect listener performance compared to the baseline but does bring down adversary accuracy, suggesting successful covert signaling. However, adding more than 2 images begins to hurt listener adversary, since the task becomes harder due to the increased context. Similar to the computational advantage setting, agents fail to generalize on the unseen test set.

## 4.2 Language analysis

How are listeners learning covert signaling protocols? Table 1 shows how adversarial pressure affects language effectiveness and systematicity on the *seen* test set[2]. First, adversarial training decreases the mutual information (NMI) of the messages with the ground-truth concepts, indicating that the language is less predictable with respect to the natural language color-shape concepts in the dataset. Second, adversarial training significantly decreases the number of unique messages sent, particularly in the computational advantage setting. This indicates that the coalition agents learn covert signaling by producing *fewer* messages to refer to the same number of concepts, rather than more messages. We see further evidence for this in the conditional entropy measurements: adversarial training increases $H(C \mid M)$, which again is most pronounced in the computational advantage setting, while $H(M \mid C)$ remains about the same. This concretely shows that *a single message refers to more concepts*. These trends collectively suggest that the coalition develops a shared convention as to what the more limited set of unique messages mean in different contexts.

---

[2]Because accuracy is poor for both the listener and the adversary on the unseen test set and suggests a degraded language protocol, we chose to highlight the language metrics on the seen test set. The metrics for the unseen set are available in the Appendix.

## 5 DISCUSSION

We showed that emergent communication agents can learn to covertly signal when given an advantage through additional training time or additional common ground. We also showed that the communication strategy of the agents changed under adversarial pressure; most notably, the speaker agent sent far fewer unique messages and used fewer messages to refer the same number of concepts. This indicates that the agents learn a more polysemous communication protocol to perform covert signaling, although additional explanation is required to fully explain the learned strategies.

A compelling direction for future work would be to compare the conventions developed by artificial agents to the human conventions that emerge in iterated reference games similar to those in Hawkins et al. (2022), but with an adversarial component. Furthermore, collecting such a dataset would also enable us to study if deep learning-based agents can learn to covertly signal when trained on natural language corpora. In conclusion, we hope that this work will help us to take some initial steps toward understanding the dynamics of communication in multi-party, mixed-incentive environments in both humans and artificial agents.

ACKNOWLEDGMENTS

We thank Rose Wang and the anonymous reviewers for helpful comments. JM is supported by an Open Philanthropy AI Fellowship.

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

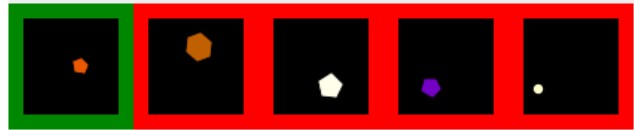

Figure 2: Example reference game. The target is indicated with a green outline.

David Kellogg Lewis. *Convention: A Philosophical Study*. Cambridge, MA, USA: Wiley-Blackwell, 1969.

Chris J. Maddison, Andriy Mnih, and Yee Whye Teh. The concrete distribution: A continuous relaxation of discrete random variables. In *5th International Conference on Learning Representations, ICLR 2017*, 2017.

John F. Nash. The bargaining problem. *Econometrica*, 18(2):155–162, 1950. ISSN 00129682, 14680262. URL http://www.jstor.org/stable/1907266.

Paul E Smaldino, Thomas J Flamson, and Richard McElreath. The evolution of covert signaling. *Scientific Reports*, 8(1), 2018.

## A  GAME DETAILS

### DATASET

The synthesized ShapeWorld dataset contains images with 10 colors and 9 shapes for a total of 90 possible concepts. The train set contains images across 72 distinct concepts (8 colors and 9 shapes). We use one test set containing only color-shape concepts seen during test time, for a total of 34 concepts. The other test set contains the 2 colors unseen during test time, and the 9 shapes, for a total of 18 novel concepts. Figure 2 shows an example of a reference game.

Following Dessí et al. (2021), we perform data augmentations on the train set inspired by the ones used in SimCLR (Chen et al., 2020), including randomized color distortion, cropping, Gaussian blurring and flipping.

In the common ground setting, the concepts depicted in the common ground images are randomly generated, meaning that they do *not* stay consistent between different example games.

### MODELS

The vision module of each agent is a 4-layer convolutional neural network, consisting of a 64-filter 3x3 convolution, batch normalization, ReLU activation function, and 2x2 max-pooling layer. RNN encoders and decoders consist of a one-layer Gated Recurrent Unit (GRU) (Cho et al., 2014) with embedding size 1024 and hidden layer size 100.

For the message encoder, we set the size of the vocabulary $|V| = 22$ and the maximum message length $k = 4$ (including a start and stop token). The reason for setting $|V| = 22$ is to encourage the models to learn concepts similar to the natural language concepts in Shapeworld, in which there are 10 colors, 9 shapes and 3 other tokens (START, STOP, PAD) for a total of 22 tokens. By similar reasoning, $k$ is set to 4 (including a START and STOP) to encourage the speaker agent to produce messages that have 1-to-1 correspondences with the natural language concepts (i.e. "START blue triangle STOP").

### TRAINING

We use the Gumbel-Softmax trick (Jang et al., 2017) with $\tau = 1$ to allow us backpropagation with categorical distributions from the discrete symbols sampled by the speaker. We optimize model parameters with the Adam optimizer (Kingma & Ba, 2015). Models are trained with learning rate 0.00005 and batch size 32, until convergence.

|  |  | $\mathrm{Acc}_L$ | $\mathrm{Acc}_A$ | $\mathrm{NMI}(M,C)$ | $n$ uniq. | $H(C \mid M)$ | $H(M \mid C)$ |
|---|---|---|---|---|---|---|---|
| **True language** | | - | - | 1 | 18 | 0 | 0 |
| **Baseline (no adversarial)** | | 0.47 | 0.44 | 0.09 | 33.8 | 3.74 | 3.81 |
| **Comp. advantage** | 1:1 | 0.10 | 0.11 | 0.18 | 89.6 | 3.28 | 3.66 |
| | 5:1 | 0.01 | 0.05 | 0.01 | 2.2 | 6.59 | 3.99 |
| | 20:1 | 0.04 | 0.09 | 0.02 | 5.4 | 6.07 | 3.96 |
| | 50:1 | 0.06 | 0.13 | 0.03 | 7.8 | 5.66 | 3.91 |
| **Common ground** | 0img | 0.10 | 0.11 | 0.18 | 89.6 | 3.28 | 3.66 |
| | 1img | 0.08 | 0.07 | 0.19 | 97.4 | 3.20 | 3.68 |
| | 2img | 0.05 | 0.05 | 0.15 | 69.8 | 3.63 | 3.70 |
| | 5img | 0.03 | 0.02 | 0.05 | 13 | 5.51 | 3.86 |

Table 2: Effects of adversarial training on metrics for *unseen* test set.

## B    LANGUAGE METRICS ON UNSEEN TEST SET

In Figure 2 we show the language effectiveness and systematicity for the *unseen* test set. The language analysis trends here are similar to those observed on the seen test set, namely that adversarial pressure decreases $\mathrm{NMI}(M,C)$, decreases the number of unique messages and increases $H(C|M)$. However, the accuracy is very low.

