# OpenReview forum: "Emergent Covert Signaling in Adversarial Reference Games"
_ICLR.cc/2022/Workshop/EmeCom — EmeCom Workshop at ICLR 2022_

### Official Review · Reviewer_E6jo · 2022-03-21
**Interesting idea that should be explored further**

**Rating:** Accept
**Confidence:** 4

**Review:**

This work proposes two agents (speaker S, listener L) cooperate to send messages in a referential game with the added twist of an adversarial eavesdropper A. The two agents aim to cooperative communicate information in such a way that the adversary cannot understand them. The game is played with a set of N images, and S must communicate which image is the true target. Results on the scenario are somewhat reasonable but generalization to an unseen test set could be improved. Overall, the authors find that increasing the learning advantage or a common knowledge between S and L (unavailable to A) is essential to do better than the adversary (despite decreasing accuracy themselves sometimes).

I find the direction of this work quite interesting (in some ways it is a discretization of Abadi and Anderson (2016)) and I believe the idea of EC for protocol creation could be a unique selling point of the field. There are some issues with the setup and the results are not very strong but I applaud the authors for their foray into a difficult setup. I believe this paper would make for great discussion and should be accepted. I leave constructive comments below so that the authors may improve their work and perhaps solve some issues.

I believe the major issue is the setup and I am not sure it is possible to achieve the desired outcome. In Abadi and Anderson, they approximate a secret-key cryptographic scenario where both S and L would have common secret knowledge unavailable to A. In contrast, the proposed setup does not have a direct parallel in cryptography. Knowledge base advantage is similar to secret key but with a few key differences: the "key" is actually distractor images that need to be feature extracted, and they are not even exactly the same due to the image augmentations used. Comp advantage doesn't make sense to me: it seems clear that artificially limiting the capabilities of A will make it worse. If the question is "under what conditions can they develop a secret protocol", it is not as interesting to answer that it is possible when the adversary is ineffective. More interesting (to me) is learning a protocol that would be secret to any adversary! It seems that with the current setup, if you continue to train A after finishing training S+L, it should learn the protocol quite well. This says to me that the setup is not one where you can learn a covert protocol. Instead, I would suggest investigating agents that share a secret key or perhaps the authors can make both agents both speakers and listeners to approximate public key. Another avenue is to compare to existing work on hiding intentions e.g. "Learning to share and hide intention by information regularization" (Strouse et al)

The secondary issue with this work is the performance. Accuracy is not incredibly high on the train set and definitely quite low on the test set. Ideally there would be some sort of baseline from another paper that would help contextualize the difficulty of the task (Dessi et al?) but in lieu of that, demonstrating high performance on a symbolic version of the dataset (color: blue, shape: square, instead of an image of a blue square) would be sufficient. I would guess that current learning has some optimization difficulties. Perhaps removing the image augmentations from Dessi et al would make the problem easier? It is also confusing why your generalization results differ in the 1:1 comp advantage vs the no-adversarial baseline. Are you updating your models following abadi and anderson, where S+L are frozen while updating A and vice-versa? Overall, the conclusions that generalization is strongly hurt by adversarial pressure seems reasonable and is intriguing but it would be an even stronger point if it could be shown proportional to the strength of the adversary.

Minor Comments
- the introduction states that EC is about solving tasks in an "unsupervised" manner but there is often supervision or at least reinforcement learning
- NMI is not really measuring systematicity but correlation between the protocol and object. TRE would be a better measure of systematicity and might be a good metric to try
- are the conditional entropies measured using the speaker and listener models? This is most likely clear but would be good to mention
- if using the gumbel-softmax trick you should also cite the concurrent work by Madison et al (2017) The Concrete Distribution
- speaker, listener, and adversary are common terms in ML but if the authors wish to approach this from a cryptographic point of view and have both agents act as speaker+listeners they can switch to Alice, Bob, and Eve (the eavesdropper)

---

### Official Review · Reviewer_8TdV · 2022-03-22
**Innovative step towards complex communication setting**

**Rating:** Accept
**Confidence:** 3

**Review:**

# Summary
This paper investigates methods to allow the emergence of a covert language in a referential game with an additional, adversarial listener. The sender-listener pair (coalition) must refer to input without the adversarial listener understanding the language. The authors then study the effect of covert communication on language structure.

They find that setting the coalition loss to minimise adversarial success is not enough for a covert language to emerge. They propose two experimental conditions where such a language can emerge : giving  more training steps to the coalition than to the adversary, or giving access to a small part of the training data exclusively to the coalition.

In these conditions, the adversary only plays the referential game at chance level, while the coalition loses a few points on a referential game without an adversarial agent, but does succeed. The language that emerges is found to be made up of fewer individual words than the setup with 2 cooperative agents, and to use the same symbols to refer to more inputs.

# Main Review
## Strengths
This paper brings emergent communication a step closer to real-world experiments, in a very simple and straightforward manner.

So far,  most work using the referential setup employed two fully cooperative agents, the main innovation of the setup presented in this paper is to find the conditions in which covert communication can emerge in a three agent setup. This is a step towards more complexity, raising potential discussion of generalisation, privacy, the sender-listener relationship… and affecting the learnt language

The contribution is simple and effectively impacts language structure. It is based on the Lewis game, very common in emergent communication, and the addition studied in this paper, the adversarial listener, is very straightforward, allowing to reproduce and build on this work.

## Weaknesses
When you describe the experimental conditions required for the emergence of a covert language it is unclear whether when your accuracy graphs are computed, both the coalition and the adversarial listener have converged, or if you are stopping after an empirical number of training steps. My worry would be that covert language emerges because the adversarial listener lacks overall training, and that it is performing as badly as any agent that has not finished learning would. Your results on language structure point towards an actual covert language :  I recommend clarifying when training is stopped.

Further along, you mention that you use data-augmentation techniques to prevent the agent focusing on low-level features. I would expect this to help the agents generalise to unseen categories, like it did in the Dessi et al 2021 paper, but this is interestingly not the case here. I would recommend clarifying what data-augmentation brings to this experiment. I’m mostly curious, did data-augmentation improve accuracy on ‘seen’ data ?

Finally it is explained in the appendix that you use 4 word messages. Multi-Symbol messages allow the study of metrics such as redundancy, or compositionality, I would look forward to reading about these effects. As far as the results of this paper are concerned, it might be interesting to check if the effects on language structure are also reachable in a simple one symbol feed-forward setup.


## Misc.

You mention that in the unskewed setup, covert languages fail to emerge. You might be interested in Cao et al (2018)’s work where adversarial communication is also seen to be an issue.

__Part 2 The Adversarial Reference Game__ contains both elements formalising the framework, and background mentioning similar works. In the final paragraph, you mention r is gradually increased while explaining a step by step run : from looking at the graph, I later understood that r was a fixed parameter that was increased in separate runs, and not dynamically during a run (am I understanding this correctly ?). I would recommend clarifying this sentence.

The concept of unseen and seen accuracy tests is mentioned in part 4.1, but are only defined in the supplementary materials. The definition should be in the main body of the article to make understanding your analysis more straightforward.

# Conclusion :
Given the results presented in this work, which show the conditions under which two cooperative agents can interact while being subject to an adversarial listener, an interesting and clear new setup with noted effects on communication, I am in favour of accepting this work and advise the authors to include the suggestions above into the camera ready.

# References :
Cao, K., Lazaridou, A., Lanctot, M., Leibo, J. Z., Tuyls, K., & Clark, S. (2018). Emergent Communication through Negotiation. International Conference on Learning Representations.

Dessì, R., Kharitonov, E., & Marco, B. (2021). Interpretable agent communication from scratch (with a generic visual processor emerging on the side). Advances in Neural Information Processing Systems, 34.

---

### Decision · Program_Chairs · 2022-03-25

**Decision:**

Accept

**Comment:**

Reviewers found the premise of this work exploring a very interesting domain. Despite some concerns about experimental setup and current lack of strong results, reviewers are excited to discuss this work and see improvements in this complex direction. We look forward to seeing it at the workshop.